# RECURRENT PARAMETER GENERATORS

## ABSTRACT

We present a generic method for recurrently using the same parameters for many different convolution layers to build a deep network. Specifically, for a network, we create a recurrent parameter generator (RPG), from which the parameters of each convolution layer are generated. Though using recurrent models to build a deep convolutional neural network (CNN) is not entirely new, our method achieves significant performance gain compared to the existing works. We demonstrate how to build a one-layer-size neural network to achieve similar performance compared to other traditional CNN models on various applications and datasets. We use the RPG to build a ResNet18 network with the number of weights equivalent to one convolutional layer of a conventional ResNet and show this model can achieve 67.2% ImageNet top-1 accuracy. Additionally, such a method allows us to build an arbitrarily complex neural network with any amount of parameters. For example, we build a ResNet34 with model parameters reduced by more than 400 times, which still achieves 41.6% ImageNet top-1 accuracy. Furthermore, the RPG can be further pruned and quantized for better run-time performance in addition to the model size reduction. We provide a new perspective for model compression. Rather than shrinking parameters from a large model, RPG sets a certain parameter-size constraint and uses the gradient descent algorithm to automatically find the best model under the constraint. Extensive experiment results are provided to demonstrate the power of the proposed recurrent parameter generator.

## 1 INTRODUCTION

Deep learning has achieved great success with increasingly more training data and deeper & larger neural networks: A recently developed NLP model, GPT-3 (Brown et al., 2020), has astonishingly 175 billion parameters! While the model performance generally scales with the number of parameters (Henighan et al., 2020), with parameters outnumbering training data, the model is significantly over-parameterized. Tremendous effort has been made to reduce the parameter redundancy from different perspectives, including neural network pruning (LeCun et al., 1990; Han et al., 2016; Liu et al., 2018), efficient network design spaces (Howard et al., 2017; Iandola et al., 2016; Sandler et al., 2018), parameter regularization (Wan et al., 2013; Wang et al., 2020a; Srivastava et al., 2014), model quantization (Hubara et al., 2017; Rastegari et al., 2016; Louizos et al., 2019), neural architecture search (Zoph & Le, 2017; Cai et al., 2018; Wan et al., 2020), recurrent models (Bai et al., 2019; 2020; Wei et al., 2016), multi-task feature encoding (Ramamonjisoa & Lepetit, 2019; Hao et al., 2021), etc.

One of the most prominent approaches in this direction is the pruning-based model compression, which dates back to the late 80s or early 90s (Mozer & Smolensky, 1989; LeCun et al., 1990) and has enjoyed a resurgence (Han et al., 2016; Blalock et al., 2020) recently. These pruning methods seek to remove the unimportant parameters from a pre-trained large neural network and can frequently achieve an enormous model-compression ratio.

Though sharing a similar motivation to reduce the parameter redundancy, we explore an entirely different territory of model parameter reduction: rather than compressing a large model, we define an arbitrarily large model based on a fixed set of parameters to maximize the model capacity. In this work, we propose to define many different layers in a deep neural network based on a fixed amount of parameters, which we call *recurrent parameter generator* (RPG). That is, we differentiate the number of model parameters and degrees of freedom (DoF). Traditionally, model parameters are treated independently of each other; the total number of parameters is the number of DoF. However, by tapping into how a core set of free parameters can be assigned to the neural network model, we can develop a large model of many parameters with a small degree of freedom. In other words,

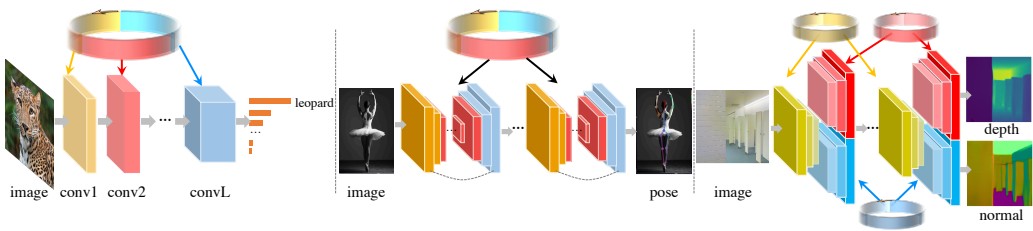

Figure 1: We propose a recurrent parameter generator (RPG) that shares a fixed set of parameters in a ring and use them to generate parameters of different parts of a neural network, whereas in the standard neural network, all the parameters are independent of each other, so the model gets bigger as it gets deeper. **Left**: The third section of the model starts to overlap with the first section in the model ring, and all later layers share generating parameters for possibly multiple times. **Right:** Employing the Recurrent Parameter Generator (RPG) for ResNet could reduce the model parameters to any size. Specifically, with only half ResNet34 backbone parameters, we achieve the same ImageNet top-1 accuracy. We also outperform model compression methods such as Knapsack (Aflalo et al., 2020).

there is excess capacity in neural network models independent of how and where the parameters are used in the network. Even at the level of individual scalar values, parameters can be reused in another arbitrary location of the deep network architecture without significantly impacting model performance. Surprisingly, backpropagation training of a deep network is able to cope with that the same parameter can be assigned to multiple random locations in the network without significantly impacting model performance. Through extensive experiments, we show that a large neural network does not need to be over overparameterized to achieve competitive performance. Particularly, a ResNet18 can be implemented with the number of weights equivalent to one convolution layer in a conventional ResNet ($4.72\times$ parameter reduction) and still achieve $67.2\%$ ImageNet top-1 accuracy. The proposed method is also extremely flexible in reducing the model parameters. In some sense, the proposed RPG method can be viewed as an automatic model parameter reduction technique, which explores the optimal accuracy-parameter trade-off. When we reduce the model parameter, RPG shows graceful performance degradation, and its compression results are frequently on par with the SOTA pruning methods besides the flexibility. Even if we reduce the ResNet18 backbone parameters to 36K, which is about $300\times$ reduction, ResNet18 can still achieve $40.0\%$ ImageNet top-1 accuracy.

Notably, we choose a destructive parameter sharing method (Cheung et al., 2019) for RPG in this work, which discourages any potential representation sharing from layer to layer. Compared to other recurrent weight-sharing methods, e.g., convolutional pose machine (CPM) or multi-scale deep equilibrium models (MDEQ), our method achieves competitive performance on various benchmarks. Further, we show RPG can be quantized and pruned to improve FLOPs and run time with very tiny accuracy drops. This makes RPG a strong and practical baseline for probing whether there is nontrivial representation sharing within any recurrent network.

To summarize, we make the following contributions:

1. This work provides a new perspective towards automatic model parameter reduction: we can define a neural network with certain DoF constraint and let gradient descent optimization automatically find the best model under the desired constraint.

Figure 2: We demonstrate the effectiveness of RPGs on various applications including image classification (**Left**), human pose estimation (**Middle**), and multitask regression (**Right**). A network can either have a global RPG or multiple local RPGs that are shared within blocks or sub-networks.

2. We propose the recurrent parameter generator (RPG), which decouples the network architecture and the network DoF. Given a certain neural network architecture, we can flexibly choose any DoF to construct the network.

3. By separating the network architecture from the parameter generator, RPG becomes a tool for us to understand the relationship between the model DoF and the network performance. We observe an empirical log-linear DoF-Accuracy relationship.

## 2 RELATED WORK

There are many important efforts to compress neural networks or to reduce the redundancy in neural network parameters. We discuss each of the approaches and their relationships to our work.

**Model Pruning, Neural Architecture Search, and Quantization.** Model pruning seeks to remove the unimportant parameters in a trained model. Recently, it's proposed to use neural architecture search as a coarse-grained model pruning (Yu et al., 2018; Dong & Yang, 2019). Another related effort is neural network quantization (Hubara et al., 2017; Rastegari et al., 2016; Louizos et al., 2019), which seeks to reduce the bits used for each parameter and can frequently reduce the model size by $4\times$ with minimal accuracy drop. More recently, Dollár et al. (2021) presents a framework for analyzing model scaling strategies that considers network properties such as FLOPs and activations.

**Parameter Regularization and Priors**. Another highly related direction is parameter regularization. Regularization has been widely used to reduce model redundancy (Krogh & Hertz, 1992), alleviate model overfitting (Srivastava et al., 2014; Wan et al., 2013), and ensure desired mathematical regularity (Wang et al., 2020a). RPG can be viewed as a parameter regularization in the sense that weight sharing poses many equality constraints to weights and regularizes weights to a low-dimensional space. HyperNeat (Stanley et al., 2009) and CPPNs (Stanley, 2007) use networks to determine the weight between two neurons as a function of their positions. Karaletsos et al. (2018) and Karaletsos & Bui (2020) introduced a similar idea by providing a hierarchical prior for network parameters.

**Recurrent Networks and Deep Equilibrium Models.** Recurrence and feedback have been shown in psychology and neuroscience to act as modulators or competitive inhibitors to aid feature grouping (Gilbert & Sigman, 2007), figure-ground segregation (Hupé et al., 1998) and object recognition (Wyatte et al., 2012). Recurrence-inspired mechanisms also achieve success in feed-forward models. There are two main types of employing recurrence based on if weights are shared across recurrent modules. ResNet (He et al., 2016), a representative of reusing similar structures without weight sharing, introduces parallel residual connections and achieves better performance by going deeper in networks. Similarly, some works (Szegedy et al., 2015; Srivastava et al., 2015) also suggest iteratively injecting thus-far representations to the feed-forward network useful. Stacked inference methods (Ramakrishna et al., 2014; Wolpert, 1992; Weiss & Taskar, 2010) are also related while they consider each output in isolation. Several works find sharing weights across recurrent modules beneficial. They demonstrate applications in temporal modelling (Weiss & Taskar, 2010; Xingjian et al., 2015; Karpathy & Fei-Fei, 2015), spatial attention (Mnih et al., 2014; Butko & Movellan, 2009), pose estimation (Wei et al., 2016; Carreira et al., 2016), and so on (Li et al., 2016; Zamir et al., 2017). Such methods usually shine in modeling long-term dependencies. In this work, we recurrently share weights across different layers of a feedback network to reduce network redundancy.

Given stacking weight-shared modules improve the performance, researchers consider running even infinite depth of such modules by making the sequential modules converge to a fixed point (LeCun et al., 1988; Bai et al., 2019). Employing such *equilibrium* models to existing networks, they show improved performance in many natural language processing (Bai et al., 2019) and computer vision tasks (Bai et al., 2020; Wang et al., 2020b). One issue with deep equilibrium models is that the forward and backward propagation usually takes much more iterations than explicit feed-forward networks. Some work (Fung et al., 2021) improves the efficiency by making the backward propagation Jacobian free. Another issue is that *infinite* depth and fixed point may not be necessary or even too strict for some tasks. Instead of achieving infinite depth, our model shares parameters to a certain level. We empirically compare with equilibrium models in Section 5.

**Efficient Network Space and Matrix Factorization.** Convolution is an efficient and structured matrix-vector multiplication. Arguably, the most fundamental idea in building efficient linear systems is matrix factorization. Given the redundancy in deep convolutional neural network parameters, one can leverage the matrix factorization concept, e.g., factorized convolutions, and design more efficient network classes (Howard et al., 2017; Iandola et al., 2016; Tan & Le, 2019; Sandler et al., 2018).

## 3 RECURRENT PARAMETER GENERATOR

We define recurrent parameter generators and show a certain kind of generating matrices that leads to destructive weight sharing. For better parameter capacity, we introduce an even sampling strategy.

**Recurrent Parameter Generator.** Assuming we are constructing a deep convolutional neural network, which contains $L$ different convolution layers. Let $\mathbf{K}_1, \mathbf{K}_2, \ldots, \mathbf{K}_L$ be the corresponding $L$ convolutional kernels [1]. Rather than using separate sets of parameters for different convolution layers, we create a single set of parameters $\mathbf{W} \in \Re^N$ and use it to generate the corresponding parameters for each convolution layer:

$$\mathbf{K}_i = \boldsymbol{R}_i \cdot \mathbf{W}, i \in \{1, \ldots, L\} \tag{1}$$

where $\boldsymbol{R}_i$ is a fixed predefined generating matrix, which is used to generate $\mathbf{K}_i$ from $\mathbf{W}$. We call $\{\boldsymbol{R}_i\}$ and $\mathbf{W}$ the *recurrent parameter generator* (RPG). In this work, we always assume that the size of $\mathbf{W}$ is smaller than the total parameters of the model, i.e., $|\mathbf{W}| \leq \sum_i |\mathbf{K}_i|$. This means an element of $\mathbf{W}$ will generally be used in more than one layer of a neural network. Additionally, the gradient of $\mathbf{W}$ is a linear superposition of the gradients from each convolution layer. During the neural network training, let's assume convolution kernel $\mathbf{K}_i$ receives gradient $\frac{\partial \ell}{\partial \mathbf{K}_i}$, where $\ell$ is the loss function. Based on the chain rule, it is clear that the gradient of $\mathbf{W}$ is:

$$\frac{\partial \ell}{\partial \mathbf{W}} = \sum_{i=1}^{L} \boldsymbol{R}_i^T \cdot \frac{\partial \ell}{\partial \mathbf{K}_i} \tag{2}$$

**Generating Matrices.** There are many different ways to create the generating matrices $\{\boldsymbol{R}_i\}$. In this work, we primarily explore the destructive generating matrices, which tend to prevent different kernels from sharing the representation during weight sharing.

**Destructive Weight Sharing.** For easier discussion, let us first look at a special case, where all of the convolutional kernels have the same size and are used in the same shape in the corresponding convolution layers. In other words, $\{\boldsymbol{R}_i\}$ are square matrices, and the spatial sizes of all of the convolutional kernels have the same size, $d_{in} \times d_{out} \times w \times h$, and the input channel dimension $d_{in}$ is always equal to the output channel dimension $d_{out}$. In this case, a filter $\mathbf{f}$ in a kernel can be treated as a vector in $\Re^{dwh}$. Further, we choose $\boldsymbol{R}_i$ to be a block-diagonal matrix $\boldsymbol{R}_i = block\text{-}diag\{\boldsymbol{A}_i, \boldsymbol{A}_i, \ldots, \boldsymbol{A}_i\}$, where $\boldsymbol{A}_i \in O(dwh)$ is an orthogonal matrix that rotates each filter from the kernel $\mathbf{K}_i$ in the same fashion. Similar to the Proposition 2 in (Cheung et al., 2019), we show in the Appendix B that: if $\boldsymbol{A}_i, \boldsymbol{A}_j$ are sampled from the $O(M)$ Haar distribution and $\mathbf{f}_i, \mathbf{f}_j$ are the same filter (generated by $\boldsymbol{R}_i, \boldsymbol{R}_j$ respectively from $\mathbf{W}$) from $\mathbf{K}_i, \mathbf{K}_j$ respectively, then we have $\mathrm{E}\left[\langle \mathbf{f}_i, \mathbf{f}_j \rangle\right] = 0$ and $\mathrm{E}\left[\langle \frac{\mathbf{f}_i}{\|\mathbf{f}_i\|}, \frac{\mathbf{f}_j}{\|\mathbf{f}_j\|} \rangle^2\right] = \frac{1}{M}$. Since $M$ is usually large, the same filter from $\mathbf{K}_i, \mathbf{K}_j$ are close to orthogonal and generally dissimilar. This shows that even when $\{\mathbf{K}_i\}$ are generated from the same $\mathbf{W}$, they are not sharing the representation.

Even though $\{\boldsymbol{A}_i\}$ are not updated during the training, the size of $\boldsymbol{A}_i$ can be quite large in general. In practice, we can use permutation $p \in P(M)$ and element-wise random sign reflection $b \in B(M)$ to construct a subset of the orthogonal group $O(M)$, i.e. we choose $\boldsymbol{A}_i \in \{b \circ p \mid b \in B(M), p \in P(M)\}$.[2] Since pseudo-random numbers are used, it takes only two random seeds to store a random permutation and an element-wise random sign reflection.

In this we work, we generalize the usage of $\boldsymbol{R}_i$ beyond the block-diagonal generating matrices described above. $\{\mathbf{K}_i\}$ may have different sizes, which can be chosen even larger than the size of $\mathbf{W}$. When $\mathbf{K}_i \in \Re^{N_i}$ is larger than $\mathbf{W} \in \Re^N$, the corresponding generating matrix $\boldsymbol{R}_i$ is a tall matrix. There are many ways to efficiently create the generating matrices. We use random permutations $P(N_i)$ and element-wise random sign reflections $B(N_i)$ to create $\boldsymbol{R}_i$:

$$R_i \in \{b \circ p \mid b \in B(N_i), p \in P(N_i)\}, \ i = 1, \ldots, L \tag{3}$$

$\{\boldsymbol{R}_i\}$ tend to lead to destructive weight sharing and lead to better utilization of the parameter capacity.

**Even Parameter Distribution for Different Layers.** While it is easy to randomly sample elements from the $\mathbf{W}$ when generating parameters for each layer, it may not be optimal as some elements in

---

[1] In this paper, we treat each convolutional kernel as a vector. When the kernel is used to do the convolution, it will be reshaped into the corresponding shape.

[2] Permutations and element-wise random sign reflection conceptually are subgroups from the orthogonal group, but we shall never use them in the matrix form for the obvious efficiency purpose.

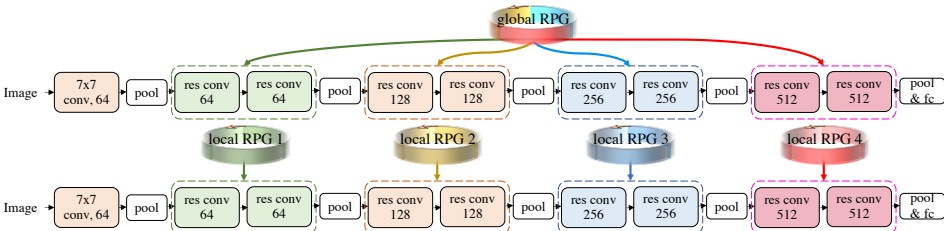

Figure 3: Recurrent parameter generators at multiple scales. **Upper:** A global RPG is used for generating convolution kernels for the entire ResNet18. **Lower**: Four local RPGs are each responsible for generating convolution kernels within each corresponding building block of the ResNet18.

$\mathbf{W}$ may never be used, and some elements may be used more than average. We use an equalization technique to guarantee all elements of $\mathbf{W}$ are evenly sampled. Suppose the size of the $\mathbf{W}$ is $N$, and the total size of parameters of layers to be generated is $M$, $M > N$. We first generate $\lfloor * \rfloor \frac{M}{N}$ arrays $\{x | x = 1, .., N\}$ and concatenate them with $(M \mod N)$ elements randomly sampled from array $\{x | x = 1, .., M\}$. We call the concatenated array index array $u \in \Re^M$. We randomly shuffle all elements in $u$. When initializing each layer's parameter, we sequentially get indices of chosen elements from the shuffled index array $u$. In this way, each layer's parameters are randomly and evenly sampled from $\mathbf{W}$. We refer to $\mathbf{W}$ as *model rings* since elements are recurrently used in a loop. For data saving efficiency, we just need to save two random seed numbers (one for sampling $(M \mod N)$ elements and one for shuffling) instead of saving the large index array $u$.

**Batch Normalization.** Model performance is relatively sensitive to the batch normalization parameters. For better performance, each of the convolution layers needs to have its own batch normalization parameters. In general, however, the size of batch normalization is relatively negligible. Yet when $\mathbf{W}$ is extremely small (e.g., 36K parameters), the size of batch normalization should be considered.

## 4 Recurrent Parameter Generator at Multiple Scales

In the previous section, we discuss the general idea of superposition where only one RPG is created and shared globally across all layers. We could also create several local RPGs, and each of them is shared at certain scales, such as blocks and sub-networks. Such super-positions may be useful for certain applications such as recurrent modeling.

**RPGs at Block-Level.** Researchers propose network architectures that reuse the same design of network blocks multiple times for higher learning capacity, as discussed in the related work. Instead of using one global RPG for the entire network, we could alternatively create several RPGs that are shared within certain network blocks. We take ResNet18 (He et al., 2016) as a concrete example (Fig.3). ResNet18 has four building blocks. Every block has 2 residual convolution modules. To superpose ResNet18 at block scale, we create four local RPGs. Each RPG is shared within the corresponding building block, where the size of the RPG is flexible and can be determined by users.

**RPGs at Sub-Network-Level.** Reusing sub-networks, or recurrent networks have achieved success in many tasks as they iteratively refine and improves the prediction. Usually, weights are shared when reusing the sub-networks. This may not be optimal as sub-networks at different stages iteratively improve the prediction, and shared weights may limit the learning capacity to adapt for different stages. On the other hand, not sharing weights at all greatly increases the model size. We superpose different sub-networks with one or more RPGs. Superposition sub-networks could have a much smaller model size, while parameters for different sub-networks undergo destructive changes instead of directly copy-paste. We show applications of superpose sub-networks for pose estimation and multitask regression (Section 5.3 and 5.4).

## 5 Experimental Results

We evaluate the performance of RPG with various tasks illustrated in Fig.2. We refer to model DoF as *number of parameters* or *parameter size* for convenience, although their differences have been discussed in Introduction. For classification, RPG was used for the entire network except for the last fully connected (fc) layer. Thus, we discuss reduction in *backbone parameters*. For example, Res18 has 11M backbone parameters and 512K fc parameters, and RPG was applied to reduce 11M backbone parameters only. Experiments are conducted on NVIDIA GeForce GTX 2080Ti GPUs.

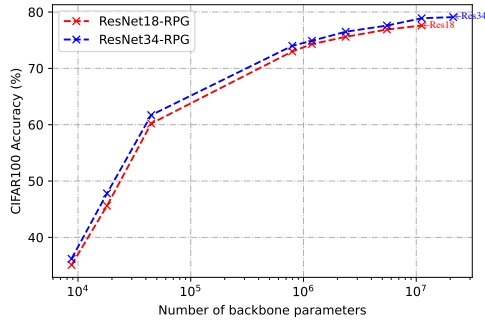
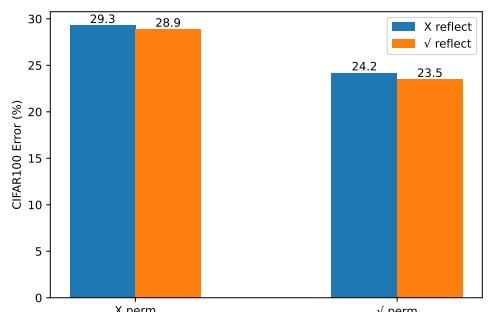

Figure 4: CIFAR100 accuracy versus the backbone parameter size for plain ResNet and RPG. RPG only has a 0.2% drop with 50% Res34 parameters.

Figure 5: Ablation studies of permutation and reflection matrices of Res34-RPG. Having both matrices gives the highest performance.

## 5.1 CIFAR CLASSIFICATION

**Implementation Details**. All CIFAR experiments use batchsize of 128, weight decay of 5e-4, and initial learning rate of 0.1 with gamma of 0.1 at epoch 60, 120 and 160. We use Kaiming initialization (He et al., 2015) with adaptive scaling. Specifically, shared parameters are initialized with a particular variance and scale the parameters for each layer to make it match the Kaiming initialization.

**Compared to Deep Equilibrium Models**. As a representative of implicit models, deep equilibrium models (Bai et al., 2019) can reduce model redundancy by finding fix points via additional optimizations. We compare the image classification accuracy on CIFAR10 and CIFAR100 datasets, as well as the inference time on CIFAR100 (Table 1). Following the settings of MDEQ (Bai et al., 2020), an image was sequentially fed into the initial convolutional block, the multi-scale deep equilibrium block (dubbed as *MS* block), and the classification head. MDEQ (Bai et al., 2020) achieves *infinite* MS blocks by finding the fixed point of the MS block. We reuse the MS block two to four times without increasing the number of parameters. Our RPG achieves 3.4% - 5.8% gain on CIFAR10 and 3% - 5.9% gain on CIFAR100. Our inference time is 15 - 25 times smaller than MDEQ since MDEQ needs additional time to solve equilibrium during training.

**Global RPG with Varying # Parameters.** We create one global RPG to generate parameters for convolution layers of ResNet and refer to it as *ResNet-RPG*. We report CIFAR100 top-1 accuracy of ResNet-RPG18 and ResNet-RPG34 at different number of parameters (Fig.4 and Table 3). Compared to ResNet, ResNet-RPG achieves higher accuracy at the same parameter size. Specifically, we achieve 36% CIFAR100 accuracy with only 8K backbone parameters. Furthermore, ResNet34-RPG achieves higher accuracy than ResNet18-RPG, indicating increasing time complexity gives performance gain.

**Local RPGs at the Block-Level**. In the previous ResNet-RPG experiments, we use one global RPG (Fig.3-**Upper**).We also evaluate the performance when RPGs are shared locally at a block level, as shown in Fig.3-**Lower**. In Table 2, compared to plain ResNet18 at the same number of parameters, our block-level RPG network gives 1.0% gain. In contrast, our ResNet-RPG (parameters are evenly distributed) gives a 1.4% gain. Using one global RPG where parameters of each layer are evenly distributed is 0.4% higher than multiple RPGs.

Table 1: Our method compared with multiscale deep equilibrium models (Bai et al., 2020) on CIFAR10 and CIFAR100 classification. At the same number of model parameters, we achieve 3% - 6% improvement with 15 - 25x less inference time. Inference time is measured by milliseconds per image.

| Accuracy (%) | MDEQ | Our RPG | | |
| --- | --- | --- | --- | --- |
| | | 2x MS blk | 3x MS blk | 4x MS blk |
| CIFAR10 | 85.1 | 88.5 | 90.1 | **90.9** |
| CIFAR100 | 59.8 | 62.8 | 64.7 | **65.7** |
| Inference time (ms) | 3.15 | **0.12** | 0.18 | 0.22 |

Table 2: At the same number of backbone parameters on CIFAR100, using local RPGs at block-level improves accuracy. Using a global RPG further improves the accuracy. RPG also outperforms baseline methods.

| | #Param | Acc. (%) |
| --- | --- | --- |
| Res18 | 11M | 77.5 |
| Res34-RPG.blk | 11M | 78.5 |
| Res34-RPG | 11M | **78.9** |
| Res34-random weight share | 11M | 74.9 |
| Res34-Hash (Chen et al., 2015) | 11M | 75.6 |
| Res34-Lego (Yang et al., 2019) | 11M | 78.4 |
| Res34 | 21M | **79.1** |

Table 3: ImageNet and CIFAR100 top-1 classification accuracy versus the number of back-bone parameters for our ResNet-RPG and plain ResNet. Our ResNet-RPG consistently achieves higher performance at the same number of parameters.

|  | ResNet18 | ResNet34 | Res18-RPG | | Res34-RPG | | |
|---|---|---|---|---|---|---|---|
| # Parameters | 11M | 21M | 45K | 2M | 45K | 2M | 11M |
| ImageNet acc. (%) | 69.8 | **73.4** | 40.0 | 67.2 | 41.6 | 69.1 | **73.4** |
| CIFAR100 acc. (%) | 77.6 | **79.1** | 60.2 | 75.6 | 61.7 | 76.5 | **78.9** |

**Comparison to Baselines.** Table 2 compares RPG and other baseline parameter reduction methods including random weight sharing, weight sharing with the hashing trick (Chen et al., 2015) and weight sharing with Lego filters Yang et al. (2019). At the same number of parameters, our RPG outperforms all other baselines, demonstrating the effectiveness of the proposed method.

## 5.2 IMAGENET CLASSIFICATION

**Implementation Details**. All ImageNet experiments use batch size of 256, weight decay of 3e-5, and an initial learning rate of 0.3 with gamma of 0.1 every 75 epochs and 225 epochs in total. Our schedule is different from the standard schedule as the weight-sharing mechanism requires different training dynamics. We tried a few settings and found this one to be the best for RPG.

**RPG with Varying # Parameters.** We use one RPG with different number of parameters for ResNet and report the top-1 accuracy (Table 3 and Fig.1(**Right**)). ResNet-RPGs consistently achieve higher performance compared to ResNets under the same number of parameters. Specifically, ResNet-RPG34 achieves the same accuracy 73.4% as ResNet34 with only half of ResNet34 backbone parameters. ResNet-RPG18 also achieves the same accuracy as ResNet18 with only half of ResNet18 backbone parameters. Further, we find RPG networks have higher generalizability (Section 5.6).

**Power Law.** Empirically, accuracy and number of parameters follow a power law, when RPG model size is lower than 50% original plain ResNet model size. The exponents of the power laws are the same for ResNet18-RPG and ResNet34-RPG on ImageNet, when comparing with ResNet34 accuracies. The scaling law may be useful for estimating the network performance without training the network. Similarly, (Henighan et al., 2020) also identifies a power law for performance and model size of transformers. The proposed RPG enables *under-parameterized* models for large-scale datasets such as ImageNet, which may unleash more new studies and findings.

## 5.3 POSE ESTIMATION

**Implementation Details.** We superpose sub-networks for pose estimation with a globally shared RPG. We use hourglass networks (Newell et al., 2016) as the building backbone. The input image is first fed to an initial convolution block to obtain a feature map. The feature map is then fed to multiple stacked pose estimation sub-networks. Each sub-network outputs a pose estimation prediction, which is penalized by the pose estimation loss. Convolutional pose machine (CPM) (Wei et al., 2016) share all the weights for different sub-networks. We create one global RPG and generate parameters for each sub-network. Our model size is set to be the same as CPM. We also compare with larger models where parameters of sub-networks are not shared.

We evaluate on MPII Human Pose dataset (Andriluka et al., 2014), a benchmark for articulated human pose estimation, which consists of over 28K training samples over 40K people with annotated body joints. We use the hourglass network (Newell et al., 2016) as backbone and follow all their settings.

**Results and Analysis.** We report the Percentage of Correct Key-points at 50% threshold (PCK@0.5) of different methods in Table 4. CPM (Wei et al., 2016) share all parameters for different sub-

Table 4: Pose estimation performance (parameter size) on MPII human pose compared with CPM (Wei et al., 2016). The metric is PCKh@0.5.

|  | CPM | Ours | No shared w. |
|---|---|---|---|
| 1x sub-net | | 84.7 (3.3M) | |
| 2x sub-nets | 86.1 (3.3M) | 86.5 (3.3M) | 87.1 (6.7M) |
| 4x sub-nets | 86.5 (3.3M) | 87.3 (3.3M) | 88.0 (13.3M) |

Table 5: Multitask regression errors on S3DIS with sub-net architecture as Ramamonjisoa & Lepetit (2019). Lower is better. Number of parameters for all methods are the same. Sub-net is reused once.

|  | Depth RMSE | Normal RMSE |
|---|---|---|
| No reusing the sub-net | 25.5% | 41.0% |
| Reuse sub-net | 24.7% | 40.3% |
| Reuse & new BN | 24.0% | 39.4% |
| Reuse & new BN & perm. and reflect. | **22.8%** | **39.1%** |

networks. We use one RPG that is shared globally at the same size as CPM. For reference, we also compare with the no-sharing model as the performance ceiling. Adding the number of recurrences leads to performance gain for all methods. At the same model size, RPG achieves higher PCK@0.5 compared to CPM. Increasing the number of parameters by not sharing sub-network parameters also leads to some performance gain.

## 5.4 MULTI-TASK REGRESSION

**Implementation Details.** We superpose sub-networks for multi-task regression with multiple RPGs at the building-block level. We focus on predicting depth and normal maps from a given image. We stack multiple SharpNet (Ramamonjisoa & Lepetit, 2019), a network for monocular depth and normal estimation. Specifically, we create multiple RPGs at the SharpNet building-block level. That is, parameters of corresponding blocks of different sub-networks are generated from the same RPG.

We evaluate the monocular depth and normal prediction performance on Standford 3D indoor scene dataset (Armeni et al., 2017). It contains over 70K images with corresponding depths and normals covering over 6,000 m$^2$ indoor area. We follow all settings of SharpNet (Ramamonjisoa & Lepetit, 2019), a state-of-the-art monocular depth and normal estimation network.

**Results and Analysis.** We report the mean square errors for depth and normal estimation in Table 5. Compared to one-time inference without recurrence, our RPG network gives 3% and 2% gain for depth and normal estimation, respectively. Directly sharing weights but using new batch normalization layers decrease the performance by 1.2% and 0.3% for depth and normal. Sharing weights and normalization layers further decrease the performance by 0.7% and 0.9% for depth and normal.

## 5.5 PRUNING RPG

Table 6: Comparison with fine-grained pruning for reducing model size. Compared with IMP (Frankle et al., 2019) on CIFAR10, RPG achieves higher pruned accuracy and similar accuracy drops.

Table 7: Comparison with coarse-grained pruning for reducing FLOPs and inference speed. RPG achieves similar performance as Knapsack (Aflalo et al., 2020) on ImageNet classification.

|  | Acc before | Acc after ↓ | Params | Acc drop | # Params |
|---|---|---|---|---|---|
| Res18-IMP | 92.3 | 90.5 | | 1.8 | 274k |
| Res18-RPG | 95.0 | 93.0 | | 2.0 | 274k |

|  | # Params before pruning | Pruned acc. | FLOPs |
|---|---|---|---|
| Res18-Knapsack | 11.2M | 69.35% | 1.09E9 |
| Pruned Res18-RPG | 5.6M | 69.10% | 1.09E9 |

**Fine-Grained Pruning**. Fine-grained pruning methods aim at reducing the model parameters by sparsifying weight matrices. Such methods usually do not reduce the inference speed, although custom algorithms (Gale et al., 2020) may improve the speed. At the same number of parameters, RPG outperforms state-of-the-art fine-grained pruning method IMP (Frankle et al., 2019). Accuracy drops of RPG and IMP are similar, both around 2% (Table 6). It is worth noting that IMP could have faster inference speed with sparse GPU kernels (Gale et al., 2020).

**Coarse-Grained Pruning**. While RPG is not designed to reduce FLOPs, it can be combined with coarse-grained pruning to reduce FLOPs. We prune RPG filters with lowest $\ell_1$ norms. Table 7 shows that the pruned RPG achieves on-par performance as state-of-the-art coarse-grained pruning method Knapsack (Aflalo et al., 2020) at the same FLOPs.

## 5.6 ANALYSIS

**Comparison to Pruning Methods**. We report our ResNet18-RPG performance with different number of parameters on ImageNet and some baseline pruning methods in Fig.1(**Right**). Our RPG networks outperform SOTA pruning methods such as (Aflalo et al., 2020; Dong & Yang, 2019; He et al., 2019; 2018; Dong et al., 2017; Khetan & Karnin, 2020). Specifically, at the same number of parameters, our RPG network has 0.6% gain over the knapsack pruning (Aflalo et al., 2020), a method that achieves the best ImageNet pruning accuracy.

**Generalizability**. We report the performance gap between training and validation set on ImageNet (Table 8(a)) and MPII pose estimation (Table 8(b)). CPM (Wei et al., 2016) serves as the baseline pose estimation method. RPG models consistently achieve lower gaps between training and validation set, indicating the RPG model suffers less from over-fitting.

We also report the out-of-distribution performance of RPG models. ObjectNet (Barbu et al., 2019) contain 50k images with 113 classes overlapping with ImageNet. Previous models are reported to

Table 8: RPG increases the model generalizability. **(a)** ResNet with RPG has the lower gap between training and validation set on ImageNet classification. The metric is training accuracy minus validation accuracy. Lower is better. **(b)** Using RPG for pose estimation also decreases the training and validation performance GAP. The metric is training PCK@0.5 minus validation PCK@0.5. Lower is better. **(c)** ResNet with RPG has higher performance on out-of-distribution dataset ObjectNet (Barbu et al., 2019). The model is trained on ImageNet only and directly evaluated on ObjectNet.

(a)

| | ResNet | RPG |
|---|---|---|
| 18 conv | -0.7 | **-2.7** |
| 34 conv | 1.1 | **-2.3** |

(b)

| | no shared w | shared w | RPG |
|---|---|---|---|
| 2x sub-nets | 1.15 | 1.13 | **0.64** |
| 4x sub-nets | 1.98 | 1.70 | **1.15** |

(c)

| | R18 | R34-RPG | R34 |
|---|---|---|---|
| # Params | 11M | 11M | 21M |
| Acc. (%) | 13.4 | **16.5** | 16.0 |

have a large performance drop on ObjectNet. We directly evaluate the performance of ImageNet-trained model on ObjectNet without any fine-tuning (Table 8(c)). With the same number of backbone parameters, our ResNet-RPG achieves a 3.1% gain compared to ResNet18. With the same network architecture design, our ResNet-RPG achieves 0.5% gain compared to ResNet34. This indicates our RPG networks have higher out-of-distribution performance even with smaller model sizes.

**Quantization.** Network quantization can reduce model size with minimal accuracy drop. It is of interest to study if RPG models, whose parameters have been shrunk, can be quantized. After 8-bit quantization, the accuracy of ResNet18-RPG (5.6M parameters) only drop 0.1 percentage point on ImageNet, indicating RPG can be quantized for further size reduction. Details are in Appendix A.

**Security**. Permutation matrices generated by the random seed can be considered as security keys to decode the model. In addition, only random seeds to generate transformation matrices need to be saved and transferred, which is efficient in terms of size.

## 5.7    ABLATION STUDIES

We conduct ablation studies to understand the functions of permutation and reflection matrices (Fig.5). We evaluate ResNet-RPG34 with 2M backbone parameters. With permutation and reflection matrices leads to 76.5% accuracy, with permutation matrices only leads to 75.8%, with reflection matrices only leads to 71.1%, and with no permutation and reflection matrices leads to 70.7%. This suggests permutation and reflection matrices are useful for RPGs.

## 6    DISCUSSION

The common practice in machine learning is to search for the best model in large model space with many parameters or degrees of freedom (DoF), and then shrink the optimal model for deployment. Our key insight is that a direct and opposite approach might work better: We start from a lean model with a small DoF, which can be unpacked into a large model with many parameters. Then we can let the gradient descent automatically find the best model under this DoF constraint. Our work is a departure from mainstream approaches towards model optimization and parameter reduction. We show how the model DoF and actual parameter size can be decoupled: We can define a large model of arbitrary number of parameters with a small DoF.

We limit our scope to linear destructive weight sharing for different convolutional layers. However, in general, there might also exist nonlinear RPGs and efficient nonlinear generation functions to create convolutional kernels from a shared model ring $\mathbf{W}$. Further, although RPG focuses on reducing model DoF, it can be quantized and pruned to further reduce the FLOPs and run time.

To sum up, we develop an efficient approach to build an arbitrarily complex neural network with any amount of DoF via a recurrent parameter generator. On a wide range of applications, including image classification, pose estimation and multitask regression, we show RPG networks consistently achieve higher performance at the same model size. Further, analysis shows that such networks are less possible to overfit and have higher performance on out-of-distribution data.

RPG can be added to any existing network flexibly with any amount of DoF at the user's discretion. It provides new perspectives for recurrent models, equilibrium models, and model compression. It also serves as a tool for understanding the relationship between network properties and network DoF by factoring out the network architecture.

**Reproducibility:** We provide our code in supplementary materials.

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
