# OpenReview forum: "Recurrent Parameter Generators"
_ICLR.cc/2022/Conference — ICLR 2022 Submitted_

### Official Review · Reviewer_cgCS · 2021-10-30

**Correctness:** 3
**Technical Novelty And Significance:** 2
**Empirical Novelty And Significance:** 2
**Recommendation:** 5
**Confidence:** 5

**Main Review:**


Strengths:
1. This paper proposes a method of weight sharing. The authors show that re-utilization of parameters generated by their recurrent parameter generator introduces diversity among kernel parameters within a single model. By reusing weights, the model size is reduced greatly.

Weaknesses:
1. This method is weak in creativity. In contrast to conventional vector quantization or just quantization methods, the RPG is not better.
2. In Section.3, they refer to the batch normalization. In the stage of inference for normal CNNs, we usually fuse the convolutional layer and batch normalization layer to one conv layer. However, the proposed method is not easy to do this. If use fusion method, extra computations will be executed.
3. In the paper "Deep compression: Compressing deep neural networks with pruning, trained quantization and huffman coding", the compression ratio is neraly 2%. I can not see any comparison in the experiments.
4. The description of "Destructive Weight Sharing" is not easy to understand.

**Summary Of The Paper:**

In order to reduce size of deep models, this work propose one parameter sharing method for different convolutional layers. With the help of one sharing set of parameters, all convolutional kernels can be generated from the sharing parameters. They show the effeciveness of this method in classification, pose estimation tasks.

**Summary Of The Review:**

Inherently, this recurrent parameter generator (RPG) is one kind of vector quantization methods, which have alread studied in some papers. Such as the citiation "LegoNet" in the paper and "Quantized Convolutional Neural Networks for Mobile Devices" in CVPR2016.

---

> ### Author Response · Authors · 2021-11-22
> **Thank you for acknowledging that the effectiveness of RPG**
>
> Thank you for acknowledging the effectiveness of RPG. We now address the concerns you raised.
>
> **[Q1]** This method is weak in creativity. Compared to conventional vector quantization or quantization methods, the RPG is not better.
>
> **[A1]**
> 1) RPG shares a similar goal with quantization in terms of parameter reduction, but the unique design and implementation make it very different from quantization methods in terms of the following: **a)** RPG decouples the network architecture and the network DoF (actual model size). Given a certain neural network architecture, we can flexibly choose any DoF to construct the network. Quantization, however, still has network architecture and size coupled and cannot easily have an arbitrary model size. **b)** By separating the network architecture from the parameter generator, RPG becomes a tool for us to understand the relationship between the model DoF and the network performance. We observe an empirical log-linear DoF-Accuracy relationship. It is difficult to use quantization methods for such studies.
> Therefore, we do not agree with the reviewer in simply saying RPG is weak in creativity just because it could reduce the model size as quantization.
> 2) We compare with quantization in the supplementary and quoted the results below. After quantization, RPG achieves a lower accuracy drop compared with vanilla Res18. We’d appreciate it if you could clarify more why “In contrast to quantization methods, RPG is not better”.
>
> | | #Params |Acc before quantization | Acc after quantization | Acc drop |
> |--|--|--|--|--|
> | Res18-vanilla|11M|69.8|69.5|0.3|
> | Res18-RPG|5.6M|70.2|70.1|0.1|
>
> **[Q2]** Normal CNNs could fuse the convolutional layer and batch normalization layer to one conv layer. It is not easy for RPG. Extra computations will be executed.
>
> **[A2]** The normalization process can be merged and incorporated into the model ring unpacking process to speed up the inference time.  After parameters of a specific layer are retrieved from the model ring, the batch normalization can be merged with the retrieved parameters before the convolution due to associativity. After this online merge, then an efficient conv-bn-relu operator can be used. We acknowledge this does introduce overhead compared to the static conv-bn merge. Let’s say the convolutional kernel size is $128 \times 128 \times 3 \times 3$; then this online merge introduces
> $128^2 \times 9$ additional multiplications in this convolution. If the output tensor size is $128 \times 20 \times 20$, then the computation overhead of this online merge is smaller than $\frac{1}{400}$ of the actual convolution since the convolution needs  $128\times128\times3\times3\times20\times20$ MACs (rather than multiplications).
>
> **[Q3]** In "Deep compression", the compression ratio is nearly 2%. Comparison in the experiments?
>
> **[A3]** Deep compression (Han et al., 2015) is a bit outdated and we compared RPG with a more recent baseline method IMP (Frankle et al., 2019) in the submission. We use the limited rebuttal time to compare with deep compression on CIFAR100 classification with ResNet18 and ResNet34 below. RPG outperforms deep compression on different number of parameters and architectures.
>
> | | R18-2M|R34-2M|R34-11M|
> |--|--|--|--|
> |deep compression|68.6|70.4|72.2|
> |RPG|75.6|76.5|78.9|
>
> **[Q4]** "Destructive Weight Sharing" is not easy to understand.
>
> **[A4]** The main goal of "Destructive Weight Sharing" is to show that with permutation and sign reflection, weights from different layers are orthogonal. Therefore, weights from different layers are decoupled to avoid too similar weights. We will revise this part to make it clearer.
>
> **[Q5]** RPG is one kind of vector quantization method, which has already been studied. Such as citations "LegoNet" and "Quantized Convolutional Neural Networks”.
>
> **[A5]** We have partially addressed this concern in [A1] in terms of quantization. Similarly for LegoNet, empirically we show RPG outperforms LegoNet. Additionally, we want to emphasize the insight of the paper, which is very different from quantization methods or LegoNet.
>
> The common practice in machine learning is to search for the best model in a large model space with many parameters or degrees of freedom (DoF), and then shrink the optimal model for deployment. Our key insight is that a direct and opposite approach might work better: We start from a lean model with a small DoF, which can be unpacked into a large model with many parameters. Then we can let the gradient descent automatically find the best model under this DoF constraint. Our work is a departure from mainstream approaches towards model optimization and parameter reduction. We show how the model DoF and actual parameter size can be decoupled: We can define a large model of an arbitrary number of parameters with a small DoF.

---

> > ### Comment · Reviewer_cgCS · 2021-11-27
> > **Response**
> >
> > Vector quantization method has already learned in "LegoNet" and "Quantized Convolutional Neural Networks for Mobile Devices". This method is the extention of this kinds of method. So I will stick to my initial judgment.

---

> > > ### Author Response · Authors · 2021-11-27
> > > **Thanks for your additional response**
> > >
> > > RPG and quantization share a different goal: RPG decouples the network architecture and the network DoF (actual model size) so we can flexibly choose any DoF to construct the network. Quantization aims to reduce the precision of network parameters for network compression. In general, RPG and quantization are connected in a way that they both can reduce # parameters of a model. However, the general connection should not block the research in understanding network properties w.r.t. DoF.

---

### Official Review · Reviewer_W9rN · 2021-11-02

**Correctness:** 3
**Technical Novelty And Significance:** 3
**Empirical Novelty And Significance:** 2
**Recommendation:** 5
**Confidence:** 5

**Details Of Ethics Concerns:**


None.


**Main Review:**



(Neutral) The parameters of each convolution layer are generated by the generator. Could the authors discuss it with "Parameter prediction for unseen deep architecture"?

(Positive) Though the novelty of this paper is limited, the proposed method achieves significant performance gain compared to the existing works.

(Negative) My main concern is that although the proposed method can build an arbitrarily complex neural network with any amount of parameters, the computational complexity is not reduced. It would be good for the authors to rethink the practical use of the proposed method. For example, although the parameters are saved, the FLOPS are not reduced. To some extent, the proposed method is related to network pruning. But all in all, the proposed technique is far less practical than network pruning methods.

(Negative) I understand that the proposed method shares some merits of convolution in weight sharing. But sharing weights in spatial is different from sharing weights in depths. Sharing weights in the spatial can (i) save parameters, (ii) can ease optimization, and (iii) can accelerate the computation by using matrix multiplication. But sharing weights in depths (i) will hurt model performances and (ii) cannot reduce computational cost. Therefore, the proposed method might not be able to be regarded as "Efficient."

(Positive) Equation (2) is elegant and precise.

(Positive) The following property is elegant: "Since M is usually large, the same filter from Ki, Kj are close to orthogonal and generally dissimilar."

(Negative) Due to the existence of matrix {A_i}, there would be additional computation cost in the generator, which further increases the computation complexity. In other words, although the parameters are saved, the computational cost might increase.

(Positive) It is reasonable to have a separate BN. It is reasonable to RPGs at Block-Level and RPGs at Sub-Network-Level.

(Negative) It would be good if the authors could apply the proposed method to the recent ViTs. Actually, based on my rich experience, vision transformers might have a more severe problem in parameter redundancy in the axis of depth. Maybe the proposed method can do well in ViTs. It would be nice if the authors could provide such studies to prove the generability of the proposed method.

(Negative) The inference time should also be added to Table 2, Table 3, and Figure 4 to show whether the proposed method is at a disadvantage.

(Positive) The observation of the Power Law is insightful, valuable, and useful, which benefits the community.

(Negative) From Table 7, we can see that even combined with the network pruning method, the proposed method does not hold an advantage, not to mention that SOTA network pruning methods are not compared with.

(Negative) Although the paper compares the proposed method with network pruning methods in Fig. 1 (right), the comparison is quite strange. Most typical network pruning methods focus on FLOPS pruning or the combination of FLOPs and parameters, but not merely parameters.


(Positive) The following result is valuable: "RPG models consistently achieve lower gaps between training and validation set, indicating the RPG model suffers less from over-fitting." The result on OOD data is also promising and valuable. It would be better to have an explanation.


**Summary Of The Paper:**


This paper uses parameter generation to generate parameters for different hidden layers in neural networks. There are several significant strengths and weaknesses in this paper. Especially, the practicability of the proposed method is questionable. It would be good to see my detailed comments below.


**Summary Of The Review:**



Balancing the strengths and weaknesses of the proposed method, I would like to recommend a rating of weak rejection for this paper unless the authors provide a persuasive response to the above negative comments.

---

> ### Author Response · Authors · 2021-11-22
> **Thanks for providing many insightful comments**
>
> Thank you for acknowledging that RPG achieves significant performance gain compared to the existing works and some properties of RPG are elegant. We now address the concerns you raised.
>
> **[Q1]** The practicability of the proposed method is questionable. The computational complexity is not reduced. The proposed method is related to network pruning. But it is far less practical than network pruning methods.
>
> **[A1]** The paper does not aim to propose a method to reduce network computation. Rather, we aim to propose RPG, which decouples the network architecture and the network DoF. By separating the network architecture from the parameter generator, RPG becomes a tool for us to understand the relationship between the model DoF and the network performance. We observe an empirical log-linear DoF-Accuracy relationship. The common practice in machine learning is to search for the best model in a large model space with many parameters or degrees of freedom (DoF), and then shrink the optimal model for deployment. Our key insight is that a direct and opposite approach might work better: We start from a lean model with a small DoF, which can be unpacked into a large model with many parameters. Then we can let the gradient descent automatically find the best model under this DoF constraint. Our work is a departure from mainstream approaches towards model optimization and parameter reduction. We show how the model DoF and actual parameter size can be decoupled: We can define a large model of an arbitrary number of parameters with a small DoF.
>
> That said, we did study the practical use of RPG by combining it with pruning methods. In Table 6, RPG achieves a similar accuracy drop after fine-grained pruning as SOTA IMP (Frankle et al., 2019). In Table 7, RPG achieves similar accuracy after coarse-grained pruning as SOTA Knapsack (Aflalo et al., 2020). RPG itself is an add-on module rather than a pruning method. We show it can be combined with pruning to achieve similar performance as SOTA pruning methods.
>
> **[Q2]** The parameters of each convolution layer are generated by the generator. Could you discuss it with "Parameter prediction for unseen deep architecture"?
>
> **[A2]** We didn’t fully understand this question. Let us try to answer this question based on our best guess. In order to use the RPG to generate a layer, the only thing we need to know is the size of the layer. So for an unseen deep architecture, we can still draw a certain size of parameters from the model ring and reshape it to the corresponding shape. As we show later in the ViT-tiny experiment that the proposed RPG method also works on ViT architecture.
>
> **[Q4]**  Sharing weights in spatial is different from sharing weights in depths. Sharing weights in the spatial can (i) save parameters, (ii) can ease optimization, and (iii) can accelerate the computation by using matrix multiplication. But sharing weights in depths (i) will hurt model performances and (ii) cannot reduce computational cost. RPG may not be "Efficient."
>
> **[A4]** We agree that compared with tiled convolution, sharing weights in spatial has many advantages. RPG has convolutional weights shared in both spatial and depth. Sharing depth does have the following advantages:
> 1) RPG could save parameters
> 2) RPG does not increase computation
> 3) Under a certain compression ratio, RPG’s performance does not go down. For example, on ImageNet, RPG-Res18 with 50% model size of vanilla-Res18 has no performance drop, and RPG-Res34 with 50% model size of vanilla-Res34 has no performance drop.
> 4) RPG can ease the optimization as the model has smaller DoF. In the table below, we show a large ViT model (86M) achieves worse performance than a small ViT-RPG model (9.3M). Also, the train-val gap is smaller for RPG. This indicates that RPG can ease the optimization by putting a constraint on the model.
>
> | |20 iter training| | | |
> |--|--|--|--|--|
> | |model size| val| train|train-val|
> |ViT-Base|86M|76.64|86.48|9.84|
> |Vit-Base-RPG|9.3M|78.95|84.54|5.59|

---

> > ### Author Response · Authors · 2021-11-22
> > **Response continued**
> >
> > **[Q5]** There would be additional computation cost in the generator, which further increases the computation complexity. The inference time should also be added.
> >
> > **[A5]** Compared with a vanilla network, the only additional computational cost or additional inference time comes from the parameter generation step.
> >
> > For a particular convolutional layer, let’s assume the convolutional kernel has $N$ parameters. The parameter generation has the following steps:
> > 1) Access the model ring to retrieve $N$ parameters.
> > 2) A permutation of the $N$ parameters.
> > 3) Flip the sign of each parameter according to the reflection array.
> >
> > Step 2) & 3) are additional compared to the baseline, where step 3) can be executed extremely efficiently. Step 2) might introduce cache misses due to the permutation operation, bounded by $2N$ memory accesses. If batch norm needs to be merged at the time of parameter generation, then this will require $N$ additional MACs.  However, these overheads are generally negligible compared to the computation of the convolutional layer. The permutation can be further made more efficient by using a hashing method to avoid the memory access overhead completely. On this overhead aspect, RPG is similar to transformer models: Their parameter generation is also relatively negligible or can be optimized to be minimal.
> >
> > **[Q6]** It would be good if the authors could apply the proposed method to the recent ViTs.
> >
> > **[A6]** Thank you for suggesting trying RPG on ViT. We use the limited rebuttal time to implement the RPG version of [ViT](https://openreview.net/pdf?id=YicbFdNTTy) on CIFAR10. In order to investigate this problem within the rebuttal time frame, we use the ViT model design space and searched for a ViT-tiny model for CIFAR10 since ViT-Base has 86M parameters and can easily overfit on CIFAR10 whereas ViT tiny has only 0.59M backbone parameters. VIT-tiny has 6 transformer layers, 4 attention heads, and 64 embedding dimensions. Overall, VIT-tiny has 0.59M backbone parameters. ViT-tiny model is small enough to train on CIFAR 10 from scratch to get a reasonable accuracy (89.1% top1).  Then we can use RPG to reduce the size. In the table below, we report ViT-tiny and ViT-tiny-RPG results with different DoF.
> >
> > | | ViT-tiny | ViT-tiny-1/2 | ViT-tiny-1/4 | ViT-tiny-1/8 | ViT-tiny-1/16 | ViT-tiny-1/32 | ViT-tiny-1/64 |
> > |---|---|---|---|---|---|---|---|
> > | model size |590K | 295K |148K |74K|37K|18K|9K|
> > | accuracy (%) | 89.1|89.0|88.4|86.0|83.1|80.0|76.5|
> >
> > We plot accuracy-DoF relationship at https://github.com/iclr22/RPG_ViT/blob/main/vit_nparam.pdf. A similar log-linear relationship is also identified in ViT.
> >
> > **[Q7]** Table 7, even combined with the network pruning method, the proposed method does not hold an advantage, not to mention that SOTA network pruning methods are not compared with.
> >
> > **[A7]**
> > 1) As we responded in [A1], we aim to project RPG as a tool that decouples the network architecture and the network DoF for many interesting studies. We do not project RPG itself as a method to reduce computation. We just show combined with pruning, RPG can reduce computation and maintain accuracy to a similar level as SOTA methods.
> > 2) To the best of our knowledge, we compare with SOTA pruning methods. For example, Knapsack (Aflalo et al., 2020) is the top-1 method for pruning on ImageNet on [paperwithcode](https://paperswithcode.com/sota/network-pruning-on-imagenet). We’d appreciate it if the reviewer could kindly point out the SOTA so we can compare.
> >
> > **[Q8]** Fig. 1 (right): the comparison is quite strange. Most typical network pruning methods focus on FLOPS pruning, but not merely parameters.
> >
> > **[A8]**
> > 1) Thank you for pointing this out. We will redo Fig.1 and its caption to make sure we mention that pruning methods also have additional FLOPs reduction.
> > 2) We did study the practical use of RPG by combining it with pruning methods. In Table 6, RPG achieves a similar accuracy drop after fine-grained pruning as SOTA IMP (Frankle et al., 2019). In Table 7, RPG achieves similar accuracy after coarse-grained pruning as SOTA Knapsack (Aflalo et al., 2020).

---

> ### Comment · Reviewer_W9rN · 2021-11-30
> **Post-rebuttal comments**
>
>
> I fully agree with reviewer HpYd's point of view at the discussion stage; that is, after reading this paper, readers will think that the main focus of this paper is model compression. This is because the motivation chapter of this paper is written like this. During the rebuttal stage, the authors emphasize that their contribution is not limited to compression but more essential. But this requires a major revision, not a minor one.
>
>
> In addition to this, my greater concern is how the methods mentioned in this paper will affect the community. On the surface, the number of parameters has been reduced, but in fact, the computational complexity has not been alleviated. The inference time should also be added to Table 2, Table 3, and Figure 4 to show whether the proposed method is at a disadvantage. But the authors did not provide the results.  I have reservations about the proposed method's usefulness and practicability. I thank the authors for providing new results on ViTs. But balancing the strengths and weaknesses of the proposed method, I think a weak rejection is better.

---

> > ### Author Response · Authors · 2021-12-01
> > **Further Discussion**
> >
> > We acknowledge that model parameter redundancy reduction is one essential motivation for this work. However, as we have made clear in the introduction, another unique motivation of RPG is to decouple the number of model parameters and degree of freedom (DoF). This dramatic difference provides a new tool to study the DoF of a neural network beyond just parameter compression. We will emphasize this more in the abstract and the introduction.
> >
> > Further, as we have discussed in detail in the rebuttal: the RPG provides a flexible DoF and parameter reduction method, and it does not reduce the computation by itself. The analysis we provide further shows that RPG only introduces negligible computational overhead. Thus, given the analysis in [A5], the inference time is a relatively unnecessary thing to show because the inference time would not change at all once we optimize the inference to deployment level. However, neural network deployment is an almost orthogonal topic to this work. That said, it would be unnecessary to show the inference time in Table 3 and Figure 4 since the inference time would not change. Regarding Table 2, ResNet 34-Hash (Chen et al. 2015) doesn’t reduce computation, so the inference time would not change much once the inference is optimized to deployment level. ResNet34-Lego (Yang et al. 2019) can reduce the computation since it is primarily a factorized-convolution method, or more specifically, it is essentially a group convolution followed by a pointwise convolution. RPG outperforms LegoNet significantly when we increase the compression ratio when it comes to redundancy reduction. We provide this simple experiment (see table below) to demonstrate the advantage of RPG - under 50x parameter reduction ratio for ResNet18, RPG achieves 67% CIFAR100 accuracy while LegoNet achieves 58%. Thus we believe the inference time would be a distraction to show in Table 2. We will include this short discussion in the revision to make this very clear.
> >
> > | | model DoF (# params) | CIFAR100 acc |
> > |--|--|--|
> > |R18-Lego| 229k|57.7%|
> > |R18-RPG| 229k|67.0%|
> >
> > *Under 50x DoF/parameter reduction ratio for ResNet18, RPG has 9 percentage point gain over LegoNet. Additionally, LegoNet cannot have arbitrary DoF as it is essentially a group convolution followed by a pointwise convolution.*

---

> > > ### Comment · Reviewer_W9rN · 2021-12-01
> > > **Further comments**
> > >
> > >
> > > Dear authors, thank you for your sincere answer. Your honest response about the inference time is in line with my expectations. In other words, the proposed method does not help reduce computational complexity. All in all, this paper is innovative, although I pay more attention to practicality. If AC and other reviewers insist that this paper should be accepted, I am also OK. Finally, I suggest that authors consider improving the paper writing. In the current version, no matter the text or the experimental design makes people feel that the primary purpose of the article is model compression.

---

### Official Review · Reviewer_HpYd · 2021-11-02

**Correctness:** 3
**Technical Novelty And Significance:** 3
**Empirical Novelty And Significance:** 3
**Recommendation:** 6
**Confidence:** 4

**Main Review:**

Pros:
The proposed technique is simple yet effective. The evaluations show that it can be applied to different types of network architectures and in different tasks. I particularly find the generalization performance valuable. While the proposed model could be expected to achieve a lower gap between the training and the validation performance due to the smaller DoF, the out-of-distribution performance improves as well.
The proposed technique also achieves better performance by using permutations of the weights in the subsequent iterations, if the base architecture follows an iterative refinement by passing the output of a layer/sub-network to itself. In my opinion, this could be a more interesting contribution claim than the parameter reduction.
The proposed work enables a more efficient parameterization by reusing a smaller set of weights. It reduces the storage space. However, the actual model size and computational complexity remain the same.

Cons and questions:
- Could the authors clarify the difference between the proposed method with the random weight assignment?  If my understanding is correct, the permutation P(M) results in a similar random assignment effect. The “even sampling” and the random sign reflections seem to be the main difference in this case. If so, why the proposed technique works significantly better than the random assignment? Unfortunately, this comparison is available only in Tab. 2 (i.e., “Res34-random weight share”). If the permutation P(M) takes away any structure in the parameter selection matrices R_i, is the model diagram in Fig.3 (upper) accurate anymore?

- This is a follow-up to my previous comment. In the pose estimation task (section 5.3), the CPM shares weights for the sub-networks and the network size remains constant as the number of sub-networks increases. RPG assigns the same weights to these sub-networks but shuffles them. Could it be concluded that using a different permutation of the same weights in the subsequent iterations improves the performance? Could the authors clarify and comment on this?
- Although parameter reduction is one of the major contribution claims, only in one experiment (Tab. 2), parameter reduction baselines (i.e., -hash and -lego entries) are used.

- It is not discussed in the paper. How does the proposed reduction technique affect the training convergence?

- The plot in Fig.1 is not clear. Only the top-right corner of the plot could be illustrated. Similarly, Fig. 4 is hard to interpret with the caption.

- I could not get the contribution claim in the security paragraph in section 5.6. Doesn’t it hold for any network?

- The following work could be discussed in the related work:
Dehghani, Mostafa, et al. "Universal transformers." arXiv preprint arXiv:1807.03819 (ICLR 19’).


-- Post-rebuttal update --

I thank the authors for their clarifications and additional experiments. After reading other reviews and responses, I decided to keep my score. As stressed by the authors, I agree that the contribution of this paper is not limited to compression. However, the experiments focus on the evaluation of the proposed method mainly in the parameter reduction tasks rather than providing insights on why and how it works. The authors addressed our concerns, yet it is still unclear why random sign flipping and even weight assignment are so effective. In my opinion, a comprehensive comparison of the proposed destructive weight-sharing approach with different concepts could improve the paper's contribution. For example, the RPG at block or sub-network levels introduces some locality. The "LegoNet" paper proposes a more structured parameter sharing approach. Comparing various techniques from fully random to slightly structured could be more insightful.
However, I still find the performance in the generalization and the OOD experiments and the applicability of the proposed approach on different architectures valuable. Hence, this paper can be considered for acceptance.

**Summary Of The Paper:**

The paper presents a technique for parameter reduction in neural networks. It follows the motivation of parameter redundancy in over-parameterized networks and aims to parameterize large networks with a smaller and shared set of weights. More specifically, a set of parameters based on a predetermined reduction factor is randomly assigned to various layers of a network and trained jointly. It is a simple technique that can be applied to any network architecture as the parameter generation is decoupled from the underlying architecture. The proposed method is evaluated in several tasks from classification to segmentation by using different network architectures. The experiments show improved or competitive performance when trained with similar or fewer parameters compared to the baselines.

**Summary Of The Review:**

The paper presents interesting results and shows that the proposed technique offers efficient parameter utilization. Applying the proposed technique in the subsequent iterations of the same network outperforms the original model. While parameter reduction is one of the main contribution claims of the paper, the experiments lack direct baselines focusing on this task.

---

> ### Author Response · Authors · 2021-11-22
> **Thank you for acknowledging that the proposed method is simple yet effective**
>
> Thank you for acknowledging that the proposed method is simple yet effective, and the generalization performance is valuable. We now address the concerns you raised.
>
> **[Q1]** Difference between the proposed method with the random weight assignment? Why RPG works significantly better than random assignments? Unfortunately, this comparison is available only in Tab. 2. If the permutation takes away any structure in the parameter selection matrices R_i, is the model diagram in Fig.3 (upper) accurate anymore?
>
> **[A1]**
> 1) Compared to random weight sharing, RPG can be considered as random and even weight assignment with random sign flipping. The improvement is not significant - on CIFAR100, random weight achieves 74.9%, and RPG achieves 78.9% (Table 2).
> 2) To clarify, in Fig.5, for ablation studies on no permutation, we sample weights of size $N$ from the first to $N^{th}$ item model ring following the original order. We will make it clearer in the revision.
> 3) We compare with random weight sharing on Res18 in the table of [A3]. We will include more comparisons in the revision.
> 4) Fig.3 (upper) is accurate.
>
> **[Q2]** In the pose estimation, the CPM shares weights for the sub-networks. RPG assigns the same weights to these sub-networks but shuffles them. Could it be concluded that using a different permutation of the same weights in the subsequent iterations improves the performance?
>
> **[A2]** Yes, and it is more precise to say that using a different permutation and sign flipping of the same weights in the subsequent iterations improves the pose estimation performance.
>
> **[Q3]** Although parameter reduction is one of the major contribution claims, only in one experiment (Tab. 2), parameter reduction baselines (i.e., -hash and -lego entries) are used.
>
> **[A3]** Thanks for the suggestion. We use the limited time to compare with parameter reduction baseline on CIFAR100 with Res18 below.
>
> | |R18-RPG|R18-rand. weight|R18-Hash|R18-Lego|
> |--|--|--|--|--|
> |Acc(%)|77.5|73.4|74.5|77.0|
>
> We will add this experiment and more comparisons to baseline methods (e.g., HyperNetworks (Ha et al., 2017) in [A2] of response to Reviewer xUeP) on more architectures and datasets in the revision.
>
> **[Q4]** How does the proposed reduction technique affect the training convergence?
>
> **[A4]** With RPG, we did not observe the training to converge faster. Specifically, as RPG can be considered as smaller models or models under constraint, we find it best to train with long epochs. For fair comparisons, all baseline methods use the same training epochs. We will make it clearer in the revision.
>
> **[Q5]** The plot in Fig.1 is not clear. Only the top-right corner of the plot could be illustrated. Similarly, Fig. 4 is hard to interpret with the caption.
>
> **[A5]** Thank you for pointing it out! We will make a clearer illustration of Fig.1 in captions. We will also redo Fig.4 and its caption in the revision.
>
> **[Q6]** I could not get the contribution claim in the security paragraph in section 5.6. Doesn’t it hold for any network?
>
> **[A6]** No, it does not hold for any network. In RPG, all parameter values are stored in the model ring. During inference, the permutation and sign reflection matrices are required to retrieve network parameters. Such matrices could be stored as random seeds or keys to *decode* the actual network parameters. For general networks, it is difficult to encode parameters to a scalar as the seed. Therefore, they do not hold the security property as RPG does.
>
> **[Q7]** The following work could be discussed: "Universal transformers." (ICLR 19’).
>
> **[A7]** Thank you for pointing this out. We will cite and discuss Universal transformers in the revision.
>
> **[Q8]** While parameter reduction is one of the main contribution claims of the paper, the experiments lack direct baselines focusing on this task.
>
> **[A8]**
> 1) As replied in [A3], we compare with parameter reduction baseline methods including random weight sharing, hashing trick (Chen et al., 2015), LegoNet (Yang et al., 2019), IMP (Frankle et al., 2019), and HyperNetworks (Ha et al., 2016). We will include more comparison results on different datasets and network architectures in the revision.
> 2) We acknowledge one main contribution of RPG lies in parameter reduction. However, we ask the reviewer to also consider two other contributions of RPG: a) it decouples the network architecture and the network DoF. Given a specific neural network architecture, we can flexibly choose any DoF to construct the network. b) By separating the network architecture from the parameter generator, RPG becomes a tool for understanding the relationship between the model DoF and the network performance. We observe an empirical log-linear DoF-Accuracy relationship.

---

> ### Author Response · Authors · 2021-12-01
> **Further Discussion on Destructive and Even Weight Sharing**
>
> Thanks for the post-rebuttal update and the positive feedback!
>
> Here we would like to provide some further discussion on weight sharing.
>
> On sign flipping: As we show in the propositions, orthogonal transformations (from the general orthogonal group) can decorrelate different filters. The intuition is that the more we decorrelate the weights, the easier it is for the network to utilize the capacity during training. In practice, to make it more efficient, we used a small subset (permutation and sign flipping) from the orthogonal group, which can be conveniently generated, and the computation overhead is minimal. If we continue to make the transformation subset smaller, it will be harder to utilize the additional capacity.
>
> On the even weight assignment: This is also to maximize the capacity utilization. If we randomly draw from the model ring, each parameter $w_j$ in the model ring would be drawn $n_j$ times. $n$ follows a relatively non-uniform distribution $P(n)$, i.e., some of the weights are used more frequently, some of them are used less or even ignored due to stochastic fluctuation. We can include this small analysis in the revision to make this clearer.

---

### Official Review · Reviewer_xUeP · 2021-11-04

**Correctness:** 3
**Technical Novelty And Significance:** 2
**Empirical Novelty And Significance:** 3
**Recommendation:** 6
**Confidence:** 3

**Main Review:**

Overall I find this work interesting, as RPG provides a simple (in the sense of how $\mathbf{K}_i$'s are generated from $\mathbf{W}$) but performant strategy of parameterizing deep neural networks.

Strengths:
1. The method itself is simple in nature, although the actual design, such as the motivation for using permutations and sign reflections, remains somewhat unclear (e.g., why not use other ways to create orthogonal matrix?). It is surprising to me how well this method works.
2. The empirical results are relatively thorough, and the improvement over baselines is substantial.

Weaknesses and questions:
1. After reading this paper I still don't understand why RPG works. For example, compared to the conventional hypernetworks (which can be considered as a sort of input-based parameter generator, and so somewhat more reasonable), what is the new and key ingredient that RPG brings? From this point of view, while the exact form of RPG is novel, using a learnable module to generate parameters **is not**. While this is an empirical paper that presents a surprising finding, I feel that some discussions of why RPG is sensible/reasonable is lacking in the current version of the paper.
2. I have some doubts over whether some experiment settings may bias towards the RPG-based networks. For instance, typically ResNets are trained on ImageNet for 100 epochs, and generally more epochs bring (diminishingly) better results. With 100 epochs, ResNet-18/-34 are actually able to reach very good performance already. While the authors acknowledge that they found "this one [setting] to be the best for RPG", I wonder if we might want to do the same for the baseline models. For instance, the numbers reported in the original ResNet paper [1]  seem already higher than in Table 3.
3. Although the method is extensively compared with other pruning or compression methods like IMP and Knapsack, the RPG method itself can also be seen as a standalone model that differs from pure compressive/pruning efforts. After all, there is nothing "compressed" or "sparse" about the modeling. The fixed generating matrix, while unlearned, is actually still **parameters of the model**, which just happens can be efficiently stored using the seed trick because of the way pseudo-random number generators work. But from a modeling point of view, the parameters $\theta$ of $f(\cdot, \theta)$ definitely still contains $\mathbf{R}_i$. This means ResNet-RPG is only **one instantiation** of the RPG framework; and I'd be very interested in seeing whether RPG works for other architectures (e.g., Transformers) that are structurally very different. Implementation wise it should be pretty straightforward; you just need a new `SuperConv1d` or `SuperLinear` module.
4. I like that the paper gets into a lot of discussions (e.g., quantization, security, log-linear DoF-accuracy relationship), but it mostly just briefly touched the surface of these topics. These are all important questions to investigate further, and the current paper hasn't been able to provide much insight other than the good empirical performance with ResNets.
5. (This is a question, not a weakness.) For the comparison with MDEQ, do you use an RPG to generate the multiscale deep equilibrium layer's parameters (which creates multi-level feature map)? Or is it also just ResNet-RPG?

Minor:
- Table 9 format is slightly messed up.
- In Sec. 5.2, the paper claims "ResNet-RPG18 ... backbone parameters". Probably you want to point of Appendix A here.


[1] https://arxiv.org/pdf/1512.03385.pdf


**Summary Of The Paper:**

The paper proposes recurrent parameter generator (RPG) that is able to generate (ideally) arbitrarily large model based on a fixed set of inputs $\mathbf{W}$. Unlike common pruning or compressing techniques, the authors argue that RPG decouples the expressivity with the degree of freedom of a model, and that we can dynamically generate model parameters on the fly (while taking advantage of the pseudo-random seed) with a simple mechanism. Experiments show that this new way of generating model parameters is able to perform on par with, or better than, many existing compressive or pruning approaches.

**Summary Of The Review:**

As mentioned in the main review, I'm not exactly satisfied with the lack of many potential discussions and further analysis of why and how RPG works. There clearly are a lot of things that can be studied here, and mere empirical evidence may not be the only thing that we need. Also, while the results are good, the method is not too different from the canonical hypernetwork idea which is also trained end-to-end. On the other hand, I feel the paper has a strong set of empirical evidence demonstrating the effectiveness of RPG and I'm surprised by the results.


-------------------

Post-rebuttal: See my comment below. I keep my score of 6.

---

> ### Author Response · Authors · 2021-11-22
> **Thank you for acknowledging that RPG provides a simple but performant strategy of parameterizing deep neural networks**
>
> Thank you for acknowledging that RPG provides a simple but performant strategy of parameterizing deep neural networks and the empirical results are relatively thorough. We now address the concerns you raised.
>
> **[Q1]** Motivation for using permutations and sign reflections is unclear (e.g., why not use other ways to create orthogonal matrix?)
>
> **[A1]** We choose permutations and sign reflections for their simplicity and negligible cost.
>
> 1) Permutations and sign reflections could be efficiently performed on hardware with tiny to no additional computation while generating orthogonal matrices usually involves computation, which may be huge given the size of the network weights.
> 2) Generation matrices need to be saved for inference. Permutation and sign reflections can be easily and efficiently saved as a random seed while saving orthogonal matrices may require space that could be even larger than the total number of network parameters. It’s certainly possible to use random matrices (from seeds) as approximate orthogonal matrices.
>
> **[Q2]** Compared to the conventional hypernetworks, what is new in RPG? The method is not too different from the canonical hypernetwork idea which is also trained end-to-end.
>
> **[A2]** This is a good point!  We actually briefly discussed the prior work HyperNeat (Stanley et al., 2009) in the initial submission.   Empirically, we find the RPG performance is much better:  Based on an implementation of HyperNetworks (https://github.com/g1910/HyperNetworks), we improve their original implementation by adding stronger augmentation such as random rotation, tuning the learning rate and schedule.  On CIFAR100 with the embedding dimension of 64 and the same model size, HyperNet has 68x FLOPs as our RPG, yet 10 percentage points lower than RPG in accuracy.
>
> | | model size (DoF) | FLOPs | CIFAR100 acc |
> |--|--|--|--|
> | HyperNet | 632K | 2.49G | 61.30% |
> | RPG | 632K | 36.7M | 71.60% |
>
> In terms of parameter/DoF reduction, RPG shares a similar goal with HyperNetworks.
>
> RPG can be considered as an extreme and minimalistic version of HyperNetworks, one without a network. However, RPG’s unique design and implementation delivers the following advantages over HyperNetworks:
> 1) HyperNetworks add substantial FLOPs to the network and render it less practical. Given a network architecture, RPG adds minimal to no additional computation, as the permutation and sign reflection can be efficiently implemented. However, HyperNetworks use a weight generation network to generate the primary network weights. A hypernet mainly uses matrix multiplication and introduces substantial FLOPs. In the table below, we analyze FLOPs of HyperNetwork for ResNet18 with the embedding dimension of 64. FLOPs of a vanilla-Res18 for ImageNet (224 input size) and CIFAR100 (32 input size) are 1.8G and 36.7M, whereas the weight generation part of the HyperNet-Res18 takes 2.45G FLOPs. This means the weight generation FLOPs are 1.4 times of vanilla-Res18 for ImageNet and 67 times of that of CIFAR100. Empirically, we find the training and inference time HyperNet-Res18 is around 70x larger than vanilla-Res18.
>
> | | ImageNet vanilla-res18 | CIFAR100 vanilla-res18 | HyperNet weight generation |
> |--|--|--|--|
> | FLOPs | 1.8G | 36.7M | 2.45G |
> | Times weight generation over | 1.36x | 66.6x | 1x |
>
> 2) HyperNetworks do not have an arbitrary DoF (number of reduced parameters). RPG uses a model ring of a size (model DoF) that can be arbitrarily determined. In HyperNetworks, the weight generation network uses the same hyper-weight and requires embedding to be of a certain size so that the matrix multiplication can be used for generating primary network weights. Therefore, the model DoF or reduced number of parameters cannot be arbitrarily determined. In other words, RPG decouples the model DoF (actual parameters) and the network architecture, while HyperNetworks have model DoF and architecture tightly coupled together, a highly restrictive limitation.
> 3) Weights generated by HyperNetworks may be coupled and not optimized for different layers. HyperNetworks use only one weight generation network parameterized by hyper-weight to generate all primary network weights. This may not be optimal as different layers of the primary network may need different weight generation networks. Additionally, matrix multiplication is used for generating weights, and the generated primary network weights may be coupled. On the other hand, RPG has destructive weight sharing, which improves the network performance by decoupling cross-layer network weights.
> We will add these results and discussions in the revision to clarify the differences between RPG and HyperNetworks.

---

> > ### Author Response · Authors · 2021-11-22
> > **Response continued**
> >
> > **[Q3]** Some experiment settings may bias towards the RPG-based networks? The numbers reported in the original ResNet paper [1] seem already higher than in Table 3.
> >
> > **[A3]** Sorry and thank you for catching a typo we made in Table 3:  Res18-vanilla (11M) should be 70.5%!  Note that Res18-RPG with 5.6M parameters also achieves 70.5%.  We accidentally reported the shorter training-schedule version. We used all the same settings for baseline methods for fair comparisons. RPG models are smaller so we adopt a longer training schedule for all methods. We will correct this typo and make it clearer in the revision.
> >
> > In the original [ResNet paper](https://arxiv.org/pdf/1512.03385), Table 2 reports the 10-crop testing performance, which is not comparable with our results as we adopt one-crop evaluation. Fig.4 of the ResNet paper reports the one-crop ImageNet performance, and by figure reading the Res18-vanilla is 68.9%  and Res34-vanilla is 72.0%. In our paper, with a longer training schedule, the Res18-vanilla and Res34-vanilla performance improved to 70.5% and 73.4%, respectively.
> >
> > **[Q4]** ResNet-RPG is only one instantiation of the RPG framework; how about RPG for other architectures (e.g., Transformers)?
> >
> > **[A4]** Thank you for suggesting trying RPG on transformers!  We use the limited rebuttal time to implement the RPG version of [ViT](https://openreview.net/pdf?id=YicbFdNTTy) on CIFAR10.  Due to time constraints, we can’t train pretrain ViT on ImageNet but train ViT from scratch instead.  However, ViT-Base with 86M parameters without pre-trained weights significantly overfit on CIFAR10. We thus search the design space and use a 590K ViT for CIFAR10. In the table below, we report ViT and ViT-RPG results with different DoF’s.
> >
> > | | ViT-vanilla | ViT-1/2 | ViT-1/4 | ViT-1/8 | ViT-1/16 | ViT-1/32 | ViT-1/64 |
> > |--|--|--|--|--|--|--|--|
> > | model size |590K | 295K |148K |74K|37K|18K|9K|
> > | accuracy (%) | 89.1|89.0|88.4|86.0|83.1|80.0|76.5|
> >
> > We plot the accuracy-DoF relationship at this anonymous link: https://github.com/iclr22/RPG_ViT/blob/main/vit_nparam.pdf. A similar log-linear relationship is also identified in ViT.
> >
> > **[Q5]** The paper gets into a lot of discussions, but mostly just briefly touched the surface of these topics.
> >
> > **[A5]** Yes, as our RPG is fundamentally different from these methods, we spend most space on presenting what it is and how it makes a difference on various tasks.  We will try to deepen our discussions in the final revision.  One example is the discussions on RPG and HyperNetworks (Please see previous discussions in [A2]).  Another example is that we would give more analysis on the log-linear accuracy-DoF relationship. A log-linear relationship has been identified in both CNNs and transformers. [Scaling Laws for Neural Language Models](https://arxiv.org/pdf/2001.08361.pdf) identifies the log-linear relationship between the test loss and parameters for language models. Since test loss and test accuracy are highly correlated, it is not surprising that we identify the log-linear relationship between accuracy and DoF.   This observation could hopefully stimulate future research from a more theoretical perspective.
> >
> > **[Q6]** For the comparison with MDEQ, do you use an RPG to generate the MDEQ layer's parameters?
> >
> > **[A6]** We use RPG to generate the multiscale deep equilibrium layer's parameters. To make a fair comparison, vanilla-MDEQ and RPG-MDEQ have the same network architecture and number of parameters.
> >
> > **[Q7]** Minor issues.
> >
> > **[A7]** Thank you for pointing them out! We will fix Table 9 layout and will point to Appendix A in the corresponding part of Section 5.2 in the revision.

---

> > > ### Comment · Reviewer_xUeP · 2021-12-01
> > > **Thank you for the response!**
> > >
> > > I'd like to thank the authors for their detailed responses and new experiments added. After reading through all the other reviews and responses, I do agree with the other reviewers that the presentation of this paper can be clearer (i.e., that this is not just about model compression), and the analysis on DoF could be more comprehensive. Also, given the drastic difference between RPG (with generation matrices and sign flipping, etc.), I think the paper will benefit significantly from much more in-depth studies of why they work, and how they work. Although the authors provided ViT results on CIFAR-10 (which I understand for the limited time), for the next version of the draft (camera-ready, for example), it'd be nice to include larger-scale results and perhaps on even more architectural variants.
> > >
> > > This is a cool idea, and I intend to maintain my current score and argue for acceptance. However, there clearly remains a lot more to be added (or answered).

---

### Author Response · Authors · 2021-11-22
**General Comments**

We thank reviewers for providing many insightful comments that help us further improve this paper.

---

### Decision · Program_Chairs · 2022-01-20

**Decision:**

Reject

**Comment:**

Meta Review for Recurrent Parameter Generators

This work investigates a method for reducing the parameters of a deep CNN by having a recurrent parameter generator (RPG) produce the weights, in effect achieving this compression via parameter sharing across layers (similar to earlier works, such as the 2016 Hypernetworks paper, as discussed in between xUeP and the author during the review period). But unlike previous work, this work conducts extensive empirical experiments on classification and even pose estimate tasks, and proposes an additional method, such as the use of pseudo-random seed to perform element-wise random sign reflection in the weight sharing. The novelty and experimental results are clearly displayed in this work, and shows a lot of promise, but after much discussion, I currently cannot recommend acceptance for ICLR 2022.

In my assessment, and also looking at reviewers and discussion, I believe this work is a great workshop paper at present, but there are a few items that would make it much stronger. There are outstanding issues in the paper that need to be improved. In particular, during discussions, reviewers noted that the paper has a problem with the design and presentation of the experiments. It somehow shifts the reader’s focus to the compression task (3 of the 4 reviewers raised concerns about the compression performance and questioned the baselines). In their rebuttal, the authors emphasized that their contribution is not limited to compression but is rather more fundamental, and the authors propose an approach for understanding the relationship between the model DoF and the network performance. But if that's the main narrative of the paper, rather than the compression aspects, the authors need to clearly articulate why decoupling the DoF from the underlying architecture is advantageous (and also make the narrative more clear in the writing). While there are novel innovations in the method proposed, the authors also need to explain clearly why their method works well, why the even weight assignment and random sign flipping are so effective?

There is discussion between the authors and reviewers about what constitutes vector quantization, and I believe the author has clarified their position effectively (with regard to cgCS's review), and I believe this will be explained in great clarity in future revisions. But even with that disagreement out of the way, we still believe that this work needs improvement to meet the bar of ICLR 2022. Reviewers, including myself, do acknowledge the novelty and are excited about the method proposed, and we look forward to seeing an updated version of this work published or presented at a future journal or conference. Good luck!